# Geographical Analysis of the Distribution of Publications Describing Spatial Associations among Outdoor Environmental Variables and Really Small Newborns in the USA and Canada

**Charlene C. Nielsen** [1,2], **Carl G. Amrhein** [3] **and Alvaro R. Osornio-Vargas** [1,2,*]

1   Department of Pediatrics, University of Alberta, Edmonton, AB T6G 1C9, Canada; ccn@ualberta.ca
2   *in*VIVO Planetary Health of the Worldwide Universities Network (WUN), West New York, NJ 07093, USA
3   Faculty of Arts and Sciences, Aga Khan University, Nairobi 00100, Kenya; carl.amrhein@aku.edu
*   Correspondence: osornio@ualberta.ca; Tel.: +1-(780)-492-7092

**Abstract:** Newborns defined as being of "low birth weight" (LBW) or "small for gestational age" (SGA) are global health issues of concern because they are vulnerable to mortality and morbidity. Prenatal exposures may contribute to LBW/SGA. In this review, we searched peer-reviewed scientific literature to determine what location-based hazards have been linked with LBW/SGA in the industrialized nations of Canada and the USA. After selecting studies based on inclusion/exclusion criteria, we entered relevant details in to an evidence table. We classified and summarized 159 articles based on type of environment (built = 108, natural = 10, and social = 41) and general category of environmental variables studied (e.g., air pollution, chemical, water contamination, waste site, agriculture, vegetation, race, SES, etc.). We linked the geographic study areas by province/state to political boundaries in a GIS to map the distributions and frequencies of the studies. We compared them to maps of LBW percentages and ubiquitous environmental hazards, including land use, industrial activity and air pollution. More studies had been completed in USA states than Canadian provinces, but the number has been increasing in both countries from 1992 to 2018. Our geographic inquiry demonstrated a novel, spatially-focused review framework to promote understanding of the human 'habitat' of shared environmental exposures that have been associated with LBW/SGA.

**Keywords:** environmental health; adverse birth outcomes; small for gestational age; low birth weight; exposome; planetary health; Canada; USA

## 1. Introduction

An underlying premise of environmental health and epidemiology involves place—where one lives and where one starts out in life, even during in utero development, ultimately determines lifelong health [1,2]. The embryo and fetus are susceptible to toxicant exposure and other environmental influences on the mother during crucial stages of pregnancy [3–6], which may lead to babies being born too small, or too early. Because they are important markers of infant survival, development, and future health, newborns that are too small are a serious source of emotional and economic stress on society—hundreds of millions of dollars are spent on specialized equipment and treatments within the first several years of life [7,8]. The Barker hypothesis [9] evolved from studies on low birth weight (as well as premature birth and intrauterine growth restriction) that found significant associations with adult hypertension, coronary heart disease, and non-insulin-dependent diabetes [10–12]. The suspected exposures associated with these birth outcomes are widespread, thus heightening the importance of early life health impacts.

The World Health Organization identifies babies born too small as an issue of global health concern, and one that is to be monitored under Sustainable Developmental Goal (SDG) 3 to "ensure healthy lives and promote wellbeing for all at all ages" (www.who.int/sdg/targets). The definitions include:

- Small for gestational age (SGA), which are infants born with a birth weight <10th percentile of a reference population for sex-based gestational age (22 to 42 weeks gestation); and
- Low birth weight at term (LBWT), which are infants born with a birth weight <2500 g, and may or may not be at full term (37–42 weeks gestation) [13–15]. Figure 1 graphically defines SGA and LBW at term.

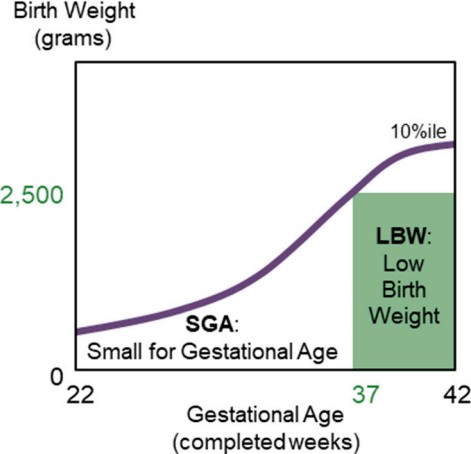

**Figure 1.** The set of birth weight–for–gestational age standards below the 10th percentile birth weights describes small for gestational age (SGA) in the purple curve; low birth weight at term (LBWT) is a subset of SGA in the green shaded rectangle.

SGA and LBWT are not homogeneous pregnancy outcomes because they may consist of both infants born too early (known as preterm birth) or too small, (typically due to fetal growth restriction) [13,16]. The etiologies are multifactorial, where the most important maternal risk factors are tobacco smoking, nutrition, pre-pregnancy weight, ethnic origin, short maternal stature, and pre-existing health conditions [16–19]. Other risks include genetic and constitutional, demographic and psychosocial (e.g., socioeconomic status (SES) and stress), obstetric, antenatal care, and toxic exposures.

Globally, the rate of SGA in low- and middle-income countries is around 27% of all live births (varying between 1.2% to 41.5% in Sahelian countries of Africa and south Asia): in 2010, 32.4 million babies were SGA [20]. LBW (all gestational ages) occurred in 15% of all births, mostly in low- and middle-income countries (mostly south Asia) [21]. Of 18 million low-birthweight babies, 10.6 million were born at term. In the United States of America (USA) in 2005, SGA was 10% [22] and LBW was 8.2% [23]. In Canada in 2005, SGA was 8.4% [24] and LBW was 6.0% [25]. Although Canada is lower than the world and U.S., disorders related to short gestation and low birth weight are consistently ranked 2nd out of the 71 leading causes of infant death [26], and their prevalence has been increasing since 2000 [24].

Figure 2 shows the geographic distribution of LBW by Canadian province and U.S. state for the years 2005 and 2016 (values for SGA unavailable). The above nationwide 2005 statistics are relevant for Figure 2a, where it can be observed that Alberta (AB), Ontario (ON), and Nunavut (NU) are higher than Canada overall, and the majority of the southern and eastern states (n = 27) are higher than USA overall.

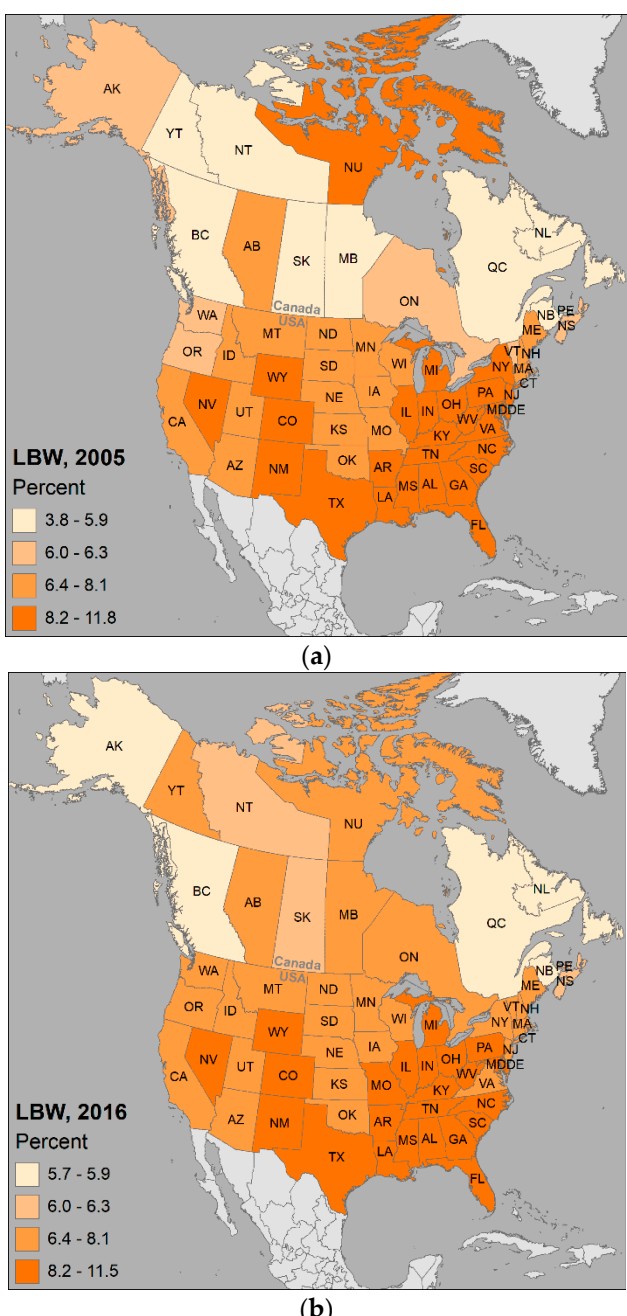

**Figure 2.** Percentage births considered low birth weight (LBW (all gestational ages)) in Canada and the USA for: (**a**) the year 2005; (**b**) the year 2016.

Given that many areas are close to or exceeding the overall national percentages, and are increasing over time as indicated by the higher number of provinces and states above 6.4 % in Figure 2, it is valuable from a public health perspective to understand the patterns and processes involved in being born too small.

SGA/LBW and their association with the environment necessitate an interdisciplinary research approach with integration of knowledge from medicine and geography. Medical geography is a holistic investigation of health using concepts and methodologies from geography, which also encompasses the social, physical, and biological sciences [27].

Informed by the earlier work of May—who stated that to understand disease as a biological expression of maladjustment, an ecological (i.e., ecosystem-based) study must involve the environment,

the host, and the culture [28]—Meade proposed the triangle of human ecology as the framework for the state of human health [27,29]. Meade's vertices are therefore anchored to:

- Habitat—the natural, social, and built environments where people live.
- Population—people (hosts) as biological organisms structured by age, gender, and genetics.
- Behavior—visible part of culture including beliefs, social organization, and technology.

These three points influence each other and the state of health, as can be seen when modelling and summarizing what is known about neonatal outcomes and maternal exposure to outdoor pollution (Figure 3). The primary population consists of pregnant mothers and their defining individual characteristics of varied ages, pre-existing health conditions and genetic makeup, with the location of where they live and work depending on their social and economic behaviors (i.e., nutritional status, access to quality health services). More research is needed that focuses on the lesser-studied habitat vertex, more specifically, the outdoor environment, since much less attention has been given to integrating ecological factors for understanding disease [27]. The location aspect of habitat (i.e., geography)—where mothers live, where industry and services are situated, where demographic groups congregate, and for many scales—is important to clinicians and specialists in environmental health, and to exposure assessment, epidemiologists, biostatisticians, and health analysts.

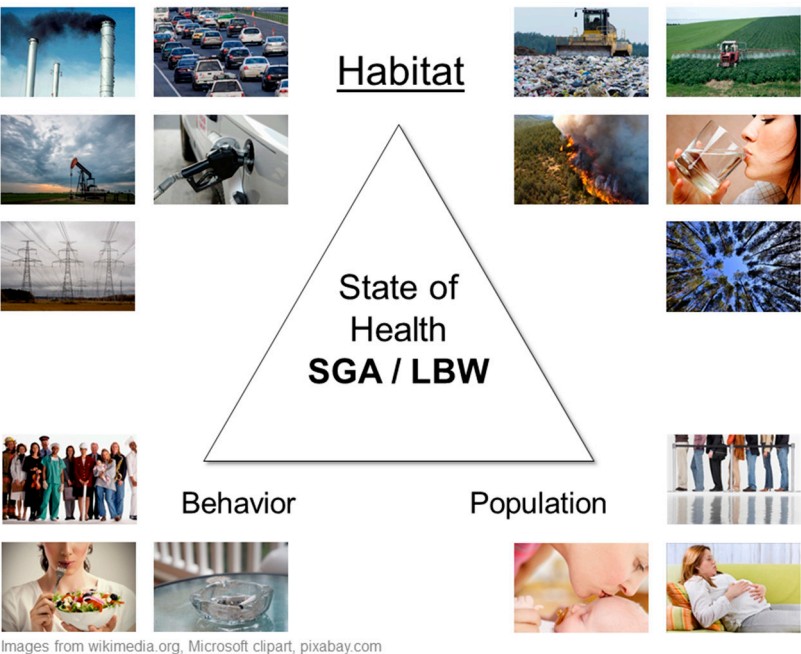

Images from wikimedia.org, Microsoft clipart, pixabay.com

**Figure 3.** Meade's triangle of human ecology for maternal exposures and small for gestational age (SGA) and low birth weight (LBW).

Geography and environmental health are inextricably linked. Environmental health, as defined by the World Health Organization, "comprises those aspects of human health and disease that are determined by factors in the environment, and includes both the direct pathological effects of chemicals, radiation and some biological agents, and the effects (often indirect) on health and wellbeing of the broad physical, psychological, social and aesthetic environment, which includes housing, urban development, land use and transport" [30]. Environmental human health is implicit in the all-encompassing planetary health, "formally defined by the invivo Planetary Health network as the interdependent vitality of all natural and anthropogenic ecosystems (social, political and otherwise)" [31,32]. These concepts are not new—Hippocrates, the father of medicine, c. 460–c. 370 BC, understood the important interconnections of environment and health, in his "Airs, Waters, and Places" [33]. Hazards in those airs, waters, and places comprise the chemical, physical, and biological

aspects that insult human health [27]. Many hazards have been known for centuries (e.g., lead, radiation, microorganisms), but they are only effective in altering health if an individual is exposed to them.

Exposure is the occurrence of a person coming into contact (via air, water, or skin) with a dose (requisite amount) of a toxicant (substance that produces a health effect) and may be isolated, repeated, or continual [34]. The health outcome can only occur if a person is exposed to the integral dose of a hazard for the crucial amount of time. These ideas are directly applicable to being born too small; the system can be simplified as follows:

$$\textit{Hazard}_{\textit{(environment)}} \rightarrow \textit{Exposure}_{\textit{(prenatal)}} \rightarrow \textit{Outcome}_{\textit{(SGA/LBW)}}$$

The measure of the total environmental exposures of an individual in a lifetime, and how those exposures relate to health, contribute to the human exposome. Evaluating the impact of the exposome is a concept of planetary health, and illuminating the exposures may contribute to understanding disease prevention [32]. This interdependence between human health and place brings us full-circle to early-life location-based exposures on pregnant mothers that may lead to really small newborns.

Mechanisms that trigger adverse birth outcomes, such as being born too small, among mothers exposed to hazards and pollutants are not well understood, but are suspected to include inflammation, direct toxic effects on the placenta and the fetus, interruption of oxygen-hemoglobin interaction, and damage to DNA [35–37]. Environmental associations differ among SGA and LBW, enhanced by temporal variations in exposures, personal characteristics (mothers' health, nutrition, and demographics) and external factors such as region and socioeconomic status (SES), [3,4,38].

Reviewing the published literature allows us to identify where information gaps exist, and also to determine whether the prevalence of the problem matches the number of existing published studies. This review serves to highlight environmental hazards, specifically, the shared exposures of the outdoor environment that have been associated with LBW and/or SGA newborns in Canada and the USA. Mapping the results will characterize where and how much LBW/SGA has been studied in the majority of industrialized North America and what and where the environmental factors are found to be important. The interested reader may use the maps as guides to what and where potential research gaps warrant further medical geographic inquiry.

## 2. Methods

### 2.1. Data Sources

Following the methodology proposed by Arksey and O'Malley [39], we searched bibliographic databases (PubMed, Web of Science, Scopus, Google Scholar, Taylor and Francis, and environmental health journal websites) to identify English-language, peer-reviewed, original research articles on outdoor environment and really small newborns. The Venn diagram in Figure 4 displays the search keywords that were used for the health outcome: (low) birth weight, small for gestational age; environmental variable: air pollution, agriculture (herbicide, pesticide, fertilizer), lead, mine, natural gas, road, traffic, (power) transmission, waste, water (contamination), socioeconomic, greenness; and any geographic extent within Canada or the USA (we read titles, abstracts and methods sections to ascertain the study country). We limited the study years to between 1990 (geographic-type analyses were rare prior) and 2018 (current year).

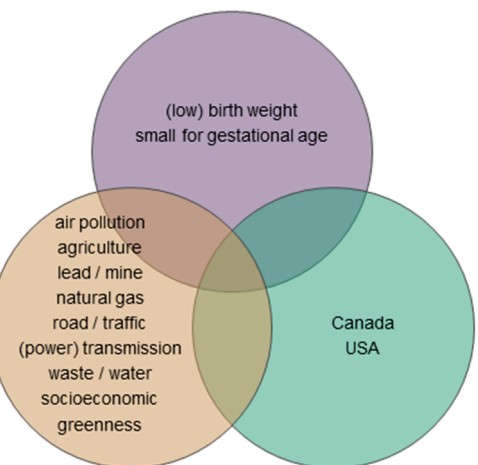

**Figure 4.** The keywords used in the literature search for studies associating outdoor environment and really small newborns are grouped by topic: health outcome (top), environmental variable (left), and geographic extent (right).

## 2.2. Study Selection and Data Extraction

We entered the articles with abstracts including both ABO and any environmental variable keywords in to Mendeley reference manager (www.mendeley.com), and tagged to identify 1 = North American and 2 = ABO. We read full articles that met the inclusion/exclusion criteria—must be Canada/USA, LBW/birth weight/SGA, and outdoor environment—and extracted the following data to a spreadsheet, formatted as the evidence table: year; study identifier; health outcome; detailed variable(s); and geography. To aid in mapping, we standardized the geography to the province or state level using the abbreviations shown in Appendix A (Table A1), regardless of whether the study was in a city, county/region, or larger administrative unit. We classified the variables in to general categories similar to the keywords, and then further generalized the environment as built, social, natural, or none. We summarized frequency statistics for the various studies. Then, we replicated records where there was more than one state or province involved in the study (e.g., a study on BC, Alberta, Manitoba, and Ontario [40] was copied to four rows in the table, one for each province) and generated a pivot table for each category or environment so that we could reliably map these for all locations.

## 2.3. Mapping

Using ArcGIS 10.6 [41], we joined the pivot table to the map of political boundaries provided by the Commission for Environmental Cooperation (CEC) [42] and created choropleth maps using four categories for the number of studies from all the selected articles—1, 2, 3, and 4 or more—labeled hereafter as frequency maps. We also mapped land use, pollution release transfer reporting (PRTR) industrial facilities [42], and satellite-based particulate matter [43]. To identify future research opportunities, these maps are compared with the 2005 and 2016 LBW percentages in Figure 2. Similarly, we visualized the frequency of studies on the built, natural, social environment, as well as those for studies related to air pollution, agriculture, chemical, vegetation, and individual factors.

## 3. Results and Discussion

The number of articles we selected for inclusion are documented in Figure 5. From the 159 included studies, associations were examined for built (n = 108), natural (n = 10), and social (n = 41) environmental variables.

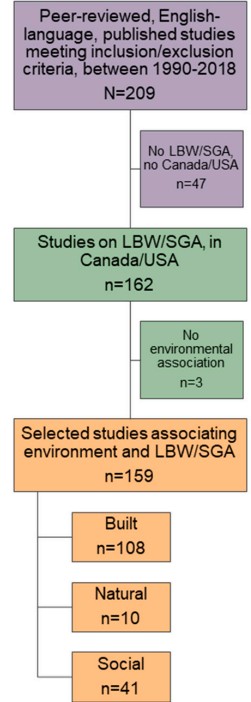

**Figure 5.** Flow diagram documenting the selection of published studies within Canada and the USA, between 1990 and 2018, for examining associations of the outdoor environment with low birth weight (LBW), birth weight (BW), and small for gestational age (SGA).

*3.1. Outcomes and Variables*

Table A2 lists all 159 studies selected for inclusion. The environmental hazards were identified as the following general categories of variables (from most-to-least frequent): air pollution (n = 53), SES (n = 17), chemical (n = 16), race (n = 11), individual (n = 10), water contamination (n = 9), waste site (n = 8), vegetation (n = 8), agriculture (n = 6), roads (n = 3), urban-rural (n = 3), food (n = 2), mining (n = 2), neighborhood (n = 2), weather (n = 2), immigration (n = 2), alcohol (n = 1), noise (n = 1), power (n = 1), transmission lines (n = 1), health care (n = 1). Note that we also included articles that studied birth weight (BW; n = 38) and intrauterine growth restriction (IUGR; n = 4) because they are interrelated with LBW (n = 72) and SGA (n = 27). There were also studies on both LBW and SGA (n = 18). Figure 6 shows how published research has increased over time from 1992 to 2018, with a peak in the year 2012. Individual states (n = 110) had more studies than Canadian provinces (n = 32), while all of USA (n = 8) and all of Canada (n = 8) were equal, with one study that included both countries.

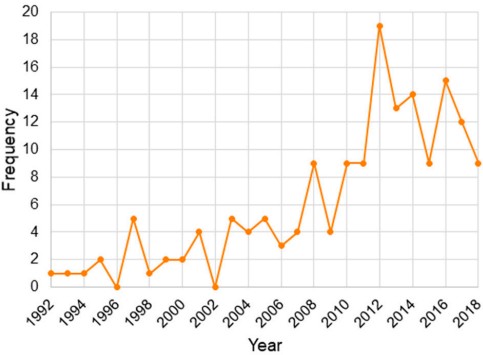

**Figure 6.** Yearly distribution of the selected studies, published on associations of the outdoor environment with low birth weight (LBW), birth weight, and small for gestational age (SGA), within Canada and the USA.

### 3.2. Spatial Associations

The following maps summarize findings from the included studies. Figure 7 maps locations and frequencies of the selected studies across North America; the distribution shows that LBW/SGA research has been conducted in six provinces and 41 states. Upon visually comparing Figure 7 with Figure 2 percentages, we observe that, despite the efforts, there are many regions with LBW and very low numbers of studies on the topic.

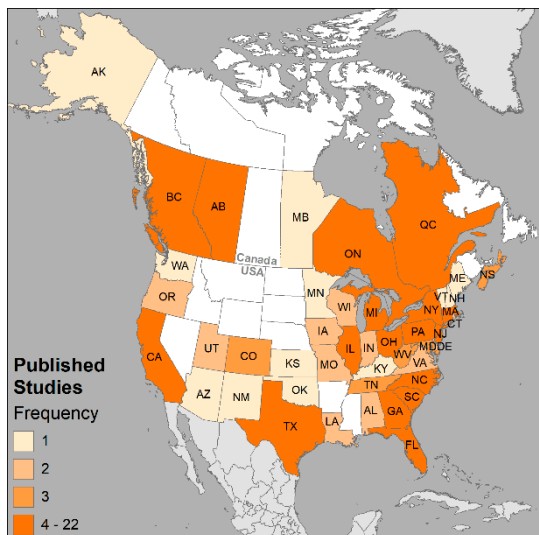

**Figure 7.** Geographic distribution and number of studies in provinces/states for the 159 candidate studies across Canada and the USA. Frequency classes standardized across all maps to intuit where the health issue is of interest (1), emerging (2), concern (3), or potential problem (4 or more).

The distributions of the types of environment (built, natural, and social) are shown in Figure 8. Figure 9 displays the most frequently studied categories.

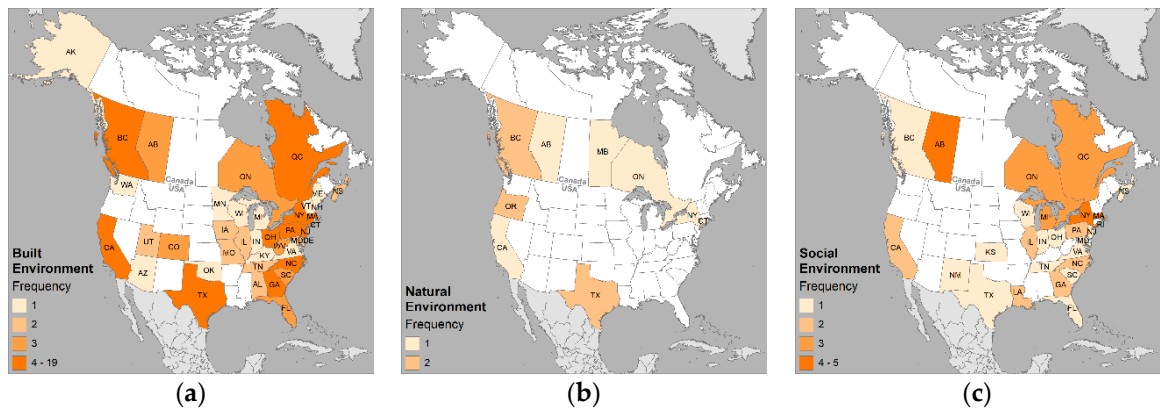

**Figure 8.** Geographic distribution of published studies by environment: (**a**) built; (**b**) natural; and (**c**) social. Frequency classes standardized across all maps to intuit where the health issue is of interest (1), emerging (2), concern (3), or potential problem (4 or more).

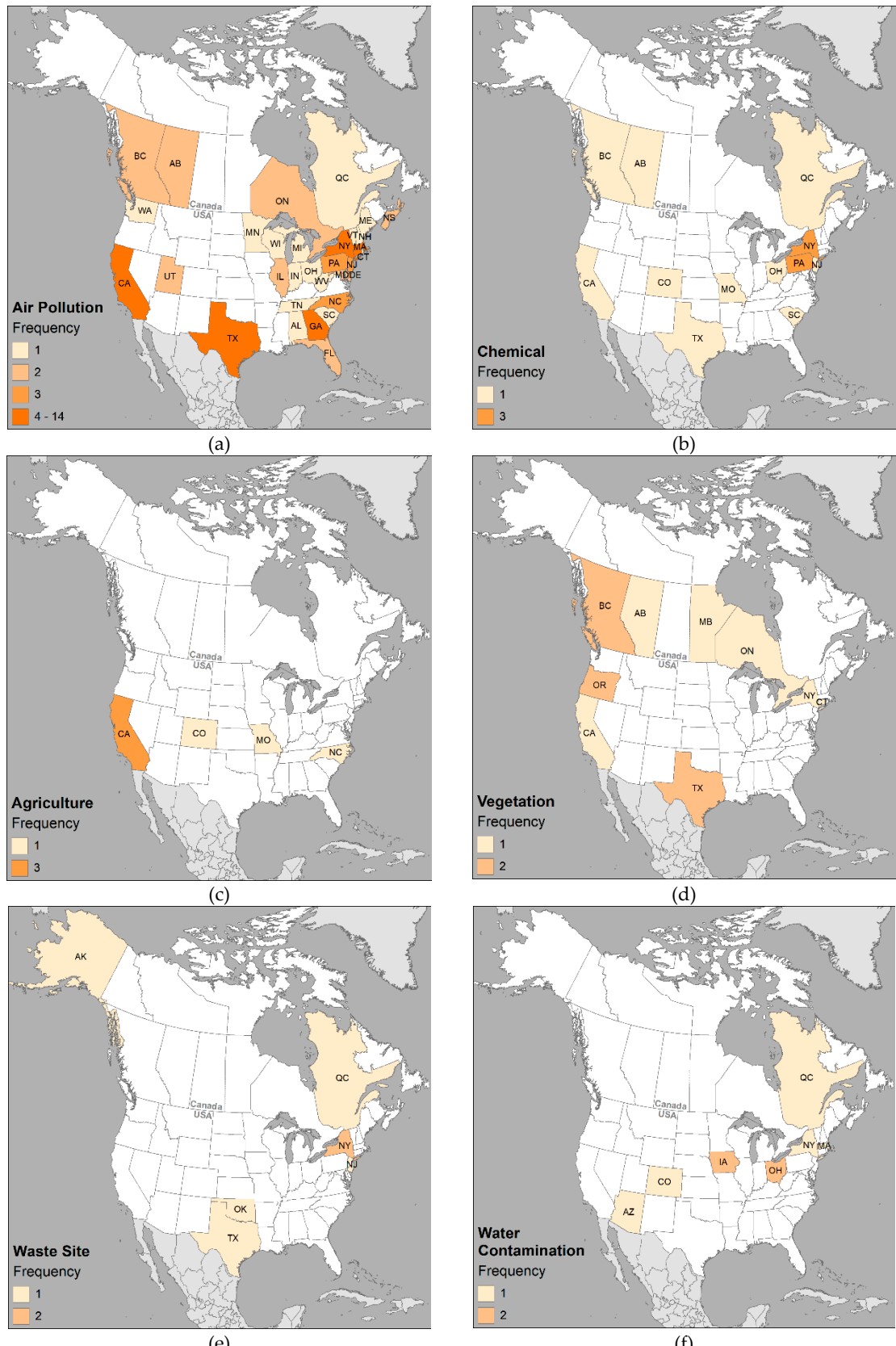

**Figure 9.** *Cont.*

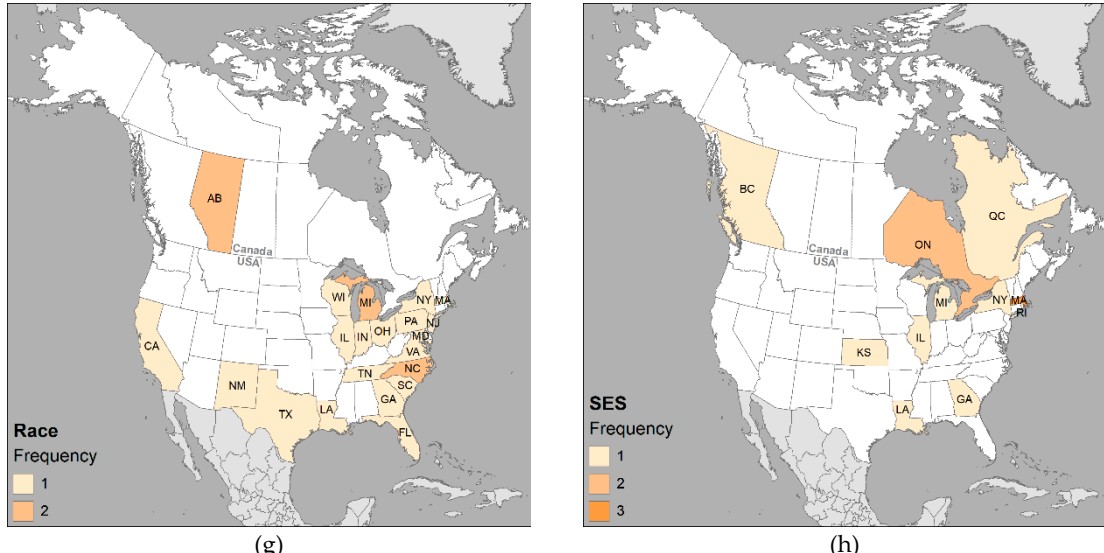

**Figure 9.** Geographic distribution of the most frequently published categories of (**a**) air pollution; (**b**) chemical; (**c**) agriculture; (**d**) vegetation; (**e**) waste site; (**f**) water contamination; (**g**) race; and (**h**) SES. Frequency classes standardized across all maps to intuit where the health issue is of interest (1), emerging (2), concern (3), or potential problem (4 or more).

For comparison purposes, the major land use classes, industrial facilities, and particulate matter distributions are mapped in Figure 10. Visual assessment highlights that the states and provinces having higher percentages of LBW in Figure 2 coincide with the same areas having relatively more proportions of urban, agriculture, industry, and PM$_{2.5}$. Inspection of the distribution of studies in Figure 7 through Figure 10 with Figure 4 shows there are clearly areas requiring future research, especially Canada's northern territories and the states bordering the Mississippi River.

### 3.3. Environmental Variables

The cumulative evidence suggested associations among outdoor environmental hazards and LBW/SGA in Canada and the USA. Most of the studies found that LBW/SGA varied with air pollution gases and/or particles depending on the trimester/gestation. Anthropogenic air pollution originates from industrial/traffic emissions and includes gaseous components—sulfur dioxide (SO$_2$), carbon monoxide (CO), nitrogen oxide (NO), nitrogen dioxide (NO$_2$), ozone (O$_3$)—and particulate matter (PM)—PM$_{2.5}$ particles with aerodynamic diameter $\leq$ 2.5 $\mu$m and PM$_{10}$ particles $\leq$ 10 $\mu$m. Electromagnetic frequencies from powerlines was not found to be important, nor was proximity to gas stations, but proximity to roads and waste sites were. The strength of association in the studies varied greatly and had limitations due to sampling, spatial resolution, availability of confounding factors, and inability to quantify duration and intensity of exposures.

Many of the previous studies linked individual or small subsets of factors; however, all factors can be modelled as vertices of the triangle of human ecology, synthesizing the complex disease ecology and advancing hypotheses [27]. As Table A2 exemplifies, the majority of air pollutants under investigation consisted of traffic-related air contaminants. A handful of studies targeted agricultural activities, heavy metals and/or industrial activities. More research is needed on assessing the spatial relationships of the actual chemicals involved in those industrial activities, especially the known or suspected developmental toxicants. Similarly, the combined effect of multipollutant exposures are still relatively unknown. Water contamination was another challenging variable, and King et al. [44] stressed the importance of household rather than distribution system sampling, making it difficult to efficiently study at a population level. Socioeconomic inequalities in LBW showed strong associations, and was larger in the United States than Canada, likely due to differing health care systems [45].

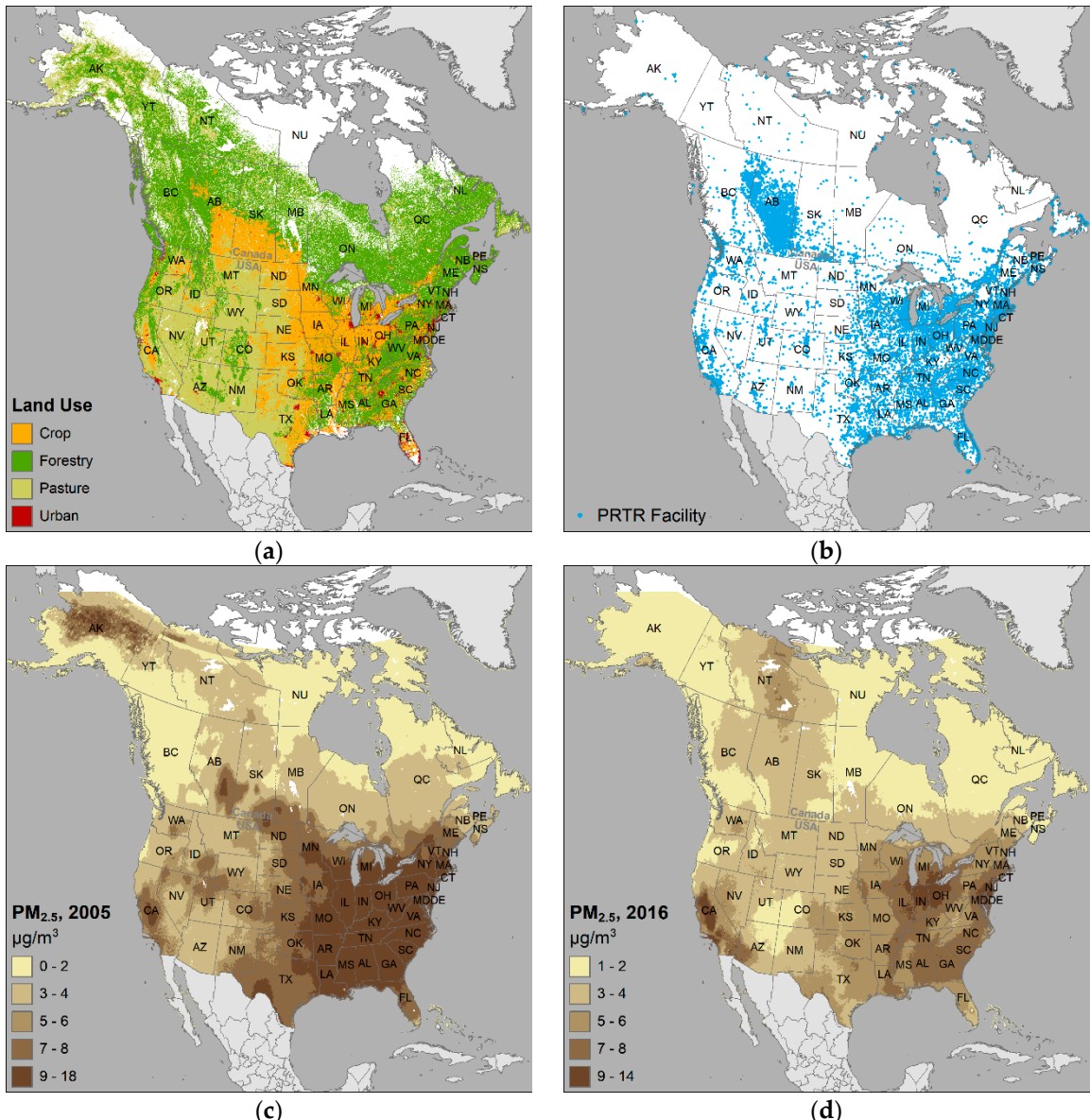

**Figure 10.** Selected environmental variables of interest in the SGA/LBW studies for Canada and USA of: (**a**) land use classes; (**b**) industrial facilities in pollutant release transfer reporting (PRTR); (**c**) common air pollutant – particulate matter particles with aerodynamic diameter $\leq 2.5$ μm, ($PM_{2.5}$) for 2005; and (**d**) $PM_{2.5}$ for 2016.

## 3.4. Exposure Assessment

Note that only English-language, peer-reviewed journal articles were selected; other literature sources have not been included here. Missing publications in other languages causes a conceptual bias, as they contribute to the overall understanding of birth weight and the environment; here, the geographic attention provides an up-to-date review on the predominantly English-publishing countries of Canada and the USA. The focus on shared sources of exposures from the outdoor environment allowed the researchers to incorporate spatial methods (i.e., GIS) in their studies, which was advantageous, especially because they facilitated several steps in exposure assessment [46]. GIS can define epidemiologic study populations, identify source and potential routes of exposure, estimate environmental levels of target contaminants, and estimate personal exposure. The studies reviewed here applied the spatial methods of coincidence, proximity, and surface predictions to identify and estimate exposures at different scales. Postal code/zip code and county-level geography

was helpful for understanding broad population patterns, but it will be worthwhile for future studies to analyze all scales with greater detail. Woodruff et al. [47] hypothesized that geographic scale was important in adverse birth outcome studies, proposing that smaller scales are useful to better understand biological mechanisms and apply to local policies, and larger scales are useful to look at population-level factors and apply to regional policy. For many of the studies, the proximity measures would benefit from increased resolution as well. An increasing number of studies are incorporating land-use regression modelling, a promising method for advancing the knowledge of exposures assessment. Analyses should also more fully integrate the socioeconomic and maternal/paternal factors, improve methods for quantifying duration and intensity of exposure, and adjust for residential mobility [35,48–50]. As previous non-spatial reviews have also stated, biological mechanisms still remain to be fully understood.

*3.5. Protective Variables*

Overall, the studies contribute to the evolving evidence that maternal exposure during pregnancy to varying levels of ambient air pollutants is associated with LBW/SGA. An interesting finding is the increase in studies on protective exposures, such as greenness—natural environments promote resiliency and prevent disease, further supporting the concept of planetary health.

## 4. Conclusions

We compiled previous spatial research on the outdoor environment and really small newborns, and through the use of maps, we presented the parameters that help with understanding how important the ambient environment is and the correspondingly valuable question of location. Such a spatially-focused review, to our knowledge, has not been seen in the literature, and we hope we have provided a useful framework for other countries to better understand environmental associations with the important global health issue of LBW and SGA newborns. North American researchers may consult these maps to aid in understanding their particular study areas.

It is hoped that our review and maps may assist healthcare professionals, in Hippocrates-style, by providing them with what location-based variables may be associated with their patients' health issues, as well as informing the public that where they live is as important to their current and future family health as what they eat and do. Our focus on environmental associations was not able to account for nutrition, maternal health, or occupation, but those studies conversely rarely accounted for outdoor exposures. Each contributes pieces to the exposome puzzle. Medical researchers are provided with more motivation for studying which components of outdoor environmental exposures may cause reduction in neonatal weight, a condition that, if prevented, will diminish future adverse health, such as adult cardiac disease, diabetes, and other non-communicable diseases that require a strong healthy start in life. Policy makers and planners (health, urban, transportation, industrial) may use this information for mitigating developments to reduce environmental effects on places where pregnant mothers (and everyone else) live. For example, existing land use may need to be altered over time depending on the proximity of industrial activities and residential areas.

May this research add to the many needed arguments for reducing the most widespread source of hazardous exposures—outdoor environmental pollution—in the places where one lives and starts out in life, to promote a more positive state of planetary health for all.

**Funding:** This research was funded by CIHR/NSERC Funding Reference Number (FRN) 127789 entitled "Spatial data mining exploring co-location of adverse birth outcomes and environmental variables."

**Acknowledgments:** Research was part of the Data Mining and Neonatal Outcomes (DoMiNO) interdisciplinary collaborative project (https://sites.google.com/a/ualberta.ca/domino/home/team-members); DoMiNO team members included Aelicks N., Aziz K., Buka I., Bellinger C., Chandra S., Demers P., Erickson A., Jabbar S., Hystad P., Kumar M., Nielsen C., Phipps E., Serrano-Lomelin J., Shah P., Stieb D., Villeneuve P., Wine O., Yuan Y., Zaiane O., and Osornio-Vargas A.

**Conflicts of Interest:** The authors declare no conflict of interest.

# Appendix A

**Table A1.** Abbreviations for Canadian provinces and USA states.

| Country | Province/State | Abbreviation |
|---|---|---|
| Canada | Alberta | AB |
| | British Columbia | BC |
| | Manitoba | MB |
| | New Brunswick | NB |
| | Newfoundland and Labrador | NL |
| | Northwest Territories | NT |
| | Nova Scotia | NS |
| | Nunavut | NU |
| | Ontario | ON |
| | Prince Edward Island | PE |
| | Quebec | QC |
| | Saskatchewan | SK |
| | Yukon Territory | YT |
| USA | Alabama | AL |
| | Alaska | AK |
| | Arizona | AZ |
| | Arkansas | AR |
| | California | CA |
| | Colorado | CO |
| | Connecticut | CT |
| | Delaware | DE |
| | District of Columbia | DC |
| | Florida | FL |
| | Georgia | GA |
| | Hawaii | HI |
| | Idaho | ID |
| | Illinois | IL |
| | Indiana | IN |
| | Iowa | IA |
| | Kansas | KS |
| | Kentucky | KY |
| | Louisiana | LA |
| | Maine | ME |
| | Maryland | MD |
| | Massachusetts | MA |
| | Michigan | MI |
| | Minnesota | MN |
| | Mississippi | MS |
| | Missouri | MO |
| | Montana | MT |
| | Nebraska | NE |
| | Nevada | NV |
| | New Hampshire | NH |
| | New Jersey | NJ |
| | New Mexico | NM |
| | New York | NY |
| | North Carolina | NC |
| | North Dakota | ND |
| | Ohio | OH |
| | Oklahoma | OK |
| | Oregon | OR |
| | Pennsylvania | PA |
| | Rhode Island | RI |
| | South Carolina | SC |
| | South Dakota | SD |

**Table A1.** *Cont.*

| Country | Province/State | Abbreviation |
|---------|----------------|--------------|
|  | Tennessee | TN |
|  | Texas | TX |
|  | Utah | UT |
|  | Vermont | VT |
|  | Virginia | VA |
|  | Washington | WA |
|  | West Virginia | WV |
|  | Wisconsin | WI |
|  | Wyoming | WY |

# Appendix B

**Table A2.** List of 159 identified studies examining birth outcomes and the environment.

| Year | Study | Outcome [1] | Environment | Category | Variable(s) | Geography [2] |
|------|-------|---------|-------------|----------|-------------|-----------|
| 2000 | Xiang et al. 2000 [51] | LBW | built | agriculture | crops | CO |
| 2010 | Fenster et al. 2010 [52] | LBW, BW | built | agriculture | agricultural occupation | CA |
| 2010 | Sathyanarayana et al. 2010 [53] | LBW | built | agriculture | pesticides | NC |
| 2013 | Gemmill et al. 2013 [54] | BW | built | agriculture | methyl bromide | CA |
| 2014 | Almberg et al. 2014 [55] | LBW | built | agriculture | crops | MO |
| 2017 | Larsen et al. 2017 [56] | BW | built | agriculture | pesticides | CA |
| 1999 | Ritz et al. 1999 [57] | LBW | built | air pollution | CO | CA |
| 2000 | Rogers et al. 2000 [58] | LBW | built | air pollution | $SO_2$, TSP | GA, SC |
| 2001 | Maisonet et al. 2001 [59] | LBW | built | air pollution | CO, $SO_2$, $PM_{10}$ | CT, MA, PA, DC |
| 2001 | Vassilev et al. 2001 [60] | SGA | built | air pollution | polycyclic organic matter | NJ |
| 2003 | Liu et al. 2003 [61] | LBW, IUGR | built | air pollution | CO, $NO_2$, $SO_2$, $O_3$, $PM_{10}$ | Canada |
| 2004 | Basu et al. 2004 [62] | BW | built | air pollution | $PM_{2.5}$ | CA |
| 2004 | Lederman et al. 2004 [63] | BW | built | air pollution | urban disaster | NY |
| 2005 | Salam et al. 2005 [64] | LBW, IUGR | built | air pollution | CO, $NO_2$, $O_3$, $PM_{10}$ | CA |
| 2006 | Dugandzic et al. 2006 [65] | LBW | built | air pollution | $PM_{10}$, $SO_2$, $O_3$ | NS |
| 2007 | Bell et al. 2007 [66] | BW | built | air pollution | CO, $NO_2$, $SO_2$, $PM_{10}$, $PM_{2.5}$ | CT, MA |
| 2007 | Liu et al. 2007 [67] | IUGR | built | air pollution | CO, $NO_2$, $SO_2$, $O_3$, $PM_{2.5}$ | AB, QC |
| 2007 | Williams et al. 2007 [68] | BW | built | air pollution | Pb, $SO_2$ | TN |
| 2008 | Brauer et al. 2008 [69] | LBW, SGA | built | air pollution | traffic | BC |
| 2008 | Choi et al. 2008 [70] | SGA | built | air pollution | PAHs | NY |
| 2009 | Currie et al. 2009 [71] | LBW | built | air pollution | industrial releases | USA |
| 2010 | Morello-Frosch et al. 2010 [72] | BW | built | air pollution | CO, $NO_2$, $SO_2$, $O_3$, $PM_{10}$, $PM_{2.5}$ | CA |
| 2011 | Darrow et al. 2011 [73] | BW | built | air pollution | CO, $NO_2$, $SO_2$, $O_3$, $PM_{10}$, $PM_{2.5}$ | GA |
| 2012 | Berrocal et al. 2012 [74] | BW | built | air pollution | $PM_{2.5}$ | NC |
| 2012 | Ebisu et al. 2012 [75] | LBW | built | air pollution | $PM_{2.5}$ | CT, DE, MD, MA, NH, NJ, NY, PA, RI, VT, VI, DC, WV |
| 2012 | Geer et al. 2012 [76] | BW | built | air pollution | CO, $NO_2$, $SO_2$, $O_3$, $PM_{10}$, $PM_{2.5}$ | TX |
| 2012 | Ghosh et al. 2012 [77] | LBW | built | air pollution | traffic | CA |
| 2012 | Holstius et al. 2012 [78] | BW | built | air pollution | wildfires | CA |
| 2012 | Kloog et al. 2012 [79] | BW | built | air pollution | $PM_{2.5}$ | MA |
| 2012 | Kumar et al. 2012 [80] | LBW | built | air pollution | CO, $NO_2$, $SO_2$, $O_3$, $PM_{10}$, $PM_{2.5}$ | IL |
| 2012 | Le et al. 2012 [81] | SGA | built | air pollution | CO, $NO_2$, $SO_2$, $O_3$, $PM_{10}$ | MI |
| 2012 | Padula et al. 2012 [82] | LBW | built | air pollution | traffic | CA |
| 2012 | Sathyanarayana et al. 2012 [83] | SGA | built | air pollution | $NO_2$, $PM_{2.5}$ | WA |

**Table A2.** *Cont.*

| Year | Study | Outcome [1] | Environment | Category | Variable(s) | Geography [2] |
|---|---|---|---|---|---|---|
| 2012 | Wilhelm et al. 2012 [84] | LBW | built | air pollution | $PM_{2.5}$, NO, $NO_2$, PAHs | CA |
| 2013 | Lee et al. 2013 [85] | SGA | built | air pollution | $PM_{10}$, $PM_{2.5}$, $O_3$ | PA |
| 2013 | Meng et al. 2013 [86] | LBW | built | air pollution | traffic | ON |
| 2013 | Trasande et al. 2013 [87] | LBW | built | air pollution | CO, $NO_2$, $SO_2$, $PM_{10}$, $PM_{2.5}$, Pb, VOCs | USA |
| 2013 | Warren et al. 2013 [88] | LBW | built | air pollution | $O_3$ | TX |
| 2014 | Basu et al. 2014 [89] | LBW | built | air pollution | $PM_{2.5}$ | CA |
| 2014 | Gray et al. 2014 [90] | BW | built | air pollution | $PM_{10}$, $PM_{2.5}$ | NC |
| 2014 | Ha et al. 2014 [91] | LBW | built | air pollution | $PM_{2.5}$, $O_3$ | FL |
| 2014 | Harris et al. 2014 [92] | LBW | built | air pollution | $PM_{2.5}$ | CT, ME, MN, NJ, NY, UT, WI |
| 2014 | Hyder et al. 2014 [93] | LBW, SGA | built | air pollution | $PM_{2.5}$ | CT, MA |
| 2014 | Porter et al. 2014 [94] | LBW | built | air pollution | industrial releases | AL |
| 2014 | Vinikoor-Imler et al. 2014 [95] | LBW, SGA | built | air pollution | $PM_{2.5}$, $O_3$ | NC |
| 2015 | Coker et al. 2015 [96] | LBW | built | air pollution | $PM_{2.5}$ | CA |
| 2015 | Poirier et al. 2015 [97] | LBW | built | air pollution | $SO_2$, $NO_2$, benzene, toluene, $PM_{10}$, $PM_{2.5}$ | NS |
| 2016 | Coker et al. 2016 [98] | LBW | built | air pollution | NO, $NO_2$, $PM_{2.5}$ | CA |
| 2016 | Erickson et al. 2016 [99] | BW | built | air pollution | $PM_{2.5}$, social | BC |
| 2016 | Laurent et al. 2016 [100] | LBW | built | air pollution | $PM_{10}$, $PM_{2.5}$ | CA |
| 2016 | Lavigne et al. 2016 [101] | LBW, SGA | built | air pollution | $PM_{2.5}$, $NO_2$, $O_3$ | ON |
| 2016 | Stieb et al. 2016 [102] | LBW, SGA, BW | built | air pollution | $NO_2$, $PM_{2.5}$ | Canada |
| 2016 | Tu et al. 2016 [103] | BW | built | air pollution | $O_3$, $PM_{2.5}$ | GA |
| 2016 | Twum et al. 2016 [104] | LBW | built | air pollution | $PM_{2.5}$ | GA |
| 2017 | Ha et al. 2017 [105] | LBW, SGA | built | air pollution | 11 criteria air contaminants and PM | CA, DC, DE, FL, UT, IL, IN, MA, MD, NY, OH, TX |
| 2017 | Jedrychowski et al. 2017 [106] | BW | built | air pollution | $PM_{2.5}$, PAH | NY |
| 2017 | Ng et al. 2017 [107] | LBW | built | air pollution | $PM_{2.5}$ | CA |
| 2017 | Nielsen et al. 2017 [108] | LBW, SGA | built | air pollution | industrial releases, built | AB |
| 2018 | Gong et al. 2018 [109] | LBW | built | air pollution | industrial releases | TX |
| 2018 | Seabrook et al. 2018 [110] | LBW | built | alcohol | alcohol | ON |
| 1992 | Shaw et al. 1992 [111] | BW | built | chemical | chemical | CA |
| 1997 | Philion et al. 1997 [112] | SGA, IUGR | built | chemical | lead | BC |
| 2004 | Lawson et al. 2004 [113] | BW | built | chemical | occupational TCDD | NJ, MO |
| 2005 | Perera et al. 2005 [114] | BW | built | chemical | ETS, PAH, pesticides | NY |
| 2008 | Wolff et al. 2008 [115] | BW | built | chemical | phenols, phthalates | NY |
| 2010 | Hamm et al. 2010 [116] | BW | built | chemical | perfluorinated acids | AB |
| 2010 | Zhu et al. 2010 [117] | BW | built | chemical | metals: Pb | NY |

**Table A2.** *Cont.*

| Year | Study | Outcome [1] | Environment | Category | Variable(s) | Geography [2] |
|---|---|---|---|---|---|---|
| 2012 | Aelion et al. 2012 [118] | BW | built | chemical | metals: As, Pb | SC |
| 2012 | Rauch et al. 2012 [119] | BW | built | chemical | pesticides | OH |
| 2014 | Mckenzie et al. 2014 [120] | LBW | built | chemical | natural gas | CO |
| 2015 | Stacy et al. 2015 [121] | SGA, BW | built | chemical | natural gas | PA |
| 2015 | Thomas et al. 2015 [122] | SGA | built | chemical | metals: Pb, Hg, Cd, As | Canada |
| 2016 | Casey et al. 2016 [123] | SGA, BW | built | chemical | natural gas | PA |
| 2017 | Whitworth et al. 2017 [124] | SGA, BW | built | chemical | natural gas | TX |
| 2018 | Ashley-Martin et al. 2018 [125] | BW | built | chemical | metals: Mn | QC |
| 2018 | Hill et al. 2018 [126] | SGA | built | chemical | natural gas | PA |
| 2008 | Lane et al. 2008 [127] | LBW | built | food | food, social | NY |
| 2016 | Ma et al. 2016 [128] | LBW, BW | built | food | food | SC |
| 2011 | Ahern et al. 2011 [129] | LBW | built | mining | coal | WV |
| 2017 | Ferdosi et al. 2017 [130] | SGA | built | mining | coal | KY, TN, VA, WV |
| 2011 | Vinikoor-Imler et al. 2011 [131] | LBW | built | neighborhood | neighborhood | NC |
| 2012 | Miranda et al. 2012 [132] | LBW, SGA | built | neighborhood | neighborhood | NC |
| 2014 | Gehring et al. 2014 [133] | LBW, BW | built | noise | noise, traffic | BC |
| 2015 | Ha et al. 2015 [134] | LBW | built | power | power plants | FL |
| 2003 | Wilhelm et al. 2003 [135] | LBW | built | roads | roads | CA |
| 2008 | Généreux et al. 2008 [136] | LBW, SGA | built | roads | roads, social | QC |
| 2012 | Miranda et al. 2012 [137] | LBW, SGA | built | roads | roads | NC |
| 2011 | Auger et al. 2011 [138] | LBW, SGA | built | transmission lines | transmission lines | QC |
| 1997 | Larson et al. 1997 [139] | LBW | built | urban-rural | urban | USA |
| 2009 | Auger et al. 2009 [140] | LBW, SGA | built | urban-rural | urban, social | QC |
| 2013 | Kent et al. 2013 [141] | LBW | built | urban-rural | urban, social | AL |
| 1994 | Sosniak et al. 1994 [142] | LBW | built | waste site | waste site | USA |
| 1995 | Goldberg et al. 1995 [143] | LBW, SGA | built | waste site | waste site | QC |
| 1997 | Berry et al. 1997 [144] | BW | built | waste site | waste site | NJ |
| 2003 | Baibergenova et al. 2003 [145] | LBW | built | waste site | waste site | NY |
| 2006 | Gilbreath et al. 2006 [146] | LBW, IUGR | built | waste site | waste site | AK |
| 2011 | Austin et al. 2011 [147] | LBW | built | waste site | waste site | NY |
| 2014 | Thompson et al. 2014 [148] | LBW | built | waste site | waste site | TX |
| 2016 | Claus et al. Henn et al. 2016 [149] | BW | built | waste site | waste site | OK |
| 1997 | Munger et al. 1997 [150] | IUGR | built | water contamination | herbicides | IA |
| 1998 | Gallagher et al. 1998 [151] | LBW | built | water contamination | trihalmethanes | CO |
| 2005 | Hinckley et al. 2005 [152] | LBW, IUGR | built | water contamination | trihalomethane, haloacetic acid | AZ |
| 2008 | Aschengrau et al. 2008 [153] | BW | built | water contamination | tetrachloroethylene | MA |

**Table A2.** *Cont.*

| Year | Study | Outcome [1] | Environment | Category | Variable(s) | Geography [2] |
|---|---|---|---|---|---|---|
| 2009 | Ochoa-Acuña et al. 2009 [154] | SGA | built | water contamination | herbicides | IA |
| 2012 | Forand et al. 2012 [155] | LBW | built | water contamination | tetrachloroethylene and trichloroethylene | NY |
| 2012 | Savitz et al. 2012 [156] | LBW, SGA | built | water contamination | perfluorooctanoic acid | OH |
| 2013 | Darrow et al. 2013 [157] | LBW, BW | built | water contamination | perfluorooctanoic acid and perfluorooctane sulfonate | OH |
| 2015 | Ileka-Priouzeau et al. 2015 [158] | SGA | built | water contamination | haloacetaldehydes, haloacetonitriles | QC |
| 2011 | Donovan et al. 2011 [159] | SGA | natural | vegetation | greenness | OR |
| 2013 | Laurent et al. 2013 [160] | BW | natural | vegetation | greenness | CA |
| 2014 | Hystad et al. 2014 [161] | SGA, BW | natural | vegetation | greenness | BC |
| 2016 | Ebisu et al. 2016 [162] | LBW, SGA, BW | natural | vegetation | greenness, built: urban | CT |
| 2017 | Abelt et al. 2017 [163] | LBW, SGA, BW | natural | vegetation | greenness, blue space | NY |
| 2017 | Cusack et al. 2017 [164] | SGA, BW | natural | vegetation | greenness | TX |
| 2017 | Cusack et al. 2017 [165] | BW | natural | vegetation | greenness | OR, TX |
| 2018 | Cusack et al. 2018 [40] | BW | natural | vegetation | greenness | BC, AB, MB, ON |
| 2012 | Lin et al. 2012 [166] | BW | natural | weather | extreme weather | USA |
| 2014 | Thayer et al. 2014 [167] | LBW | natural | weather | UV-vitamin D, social: race | USA |
| 2016 | Savard et al. 2016 [168] | SGA | social | health care | health care | QC |
| 2010 | Urquia et al. 2010 [169] | BW | social | immigration | immigration | ON |
| 2011 | Janevic et al. 2011 [170] | SGA | social | immigration | immigration | NY |
| 1995 | Mclafferty et al. 1995 [171] | LBW | social | individual | social | NY |
| 2001 | Tough et al. 2001 [172] | LBW | social | individual | maternal health | AB |
| 2003 | English et al. 2003 [173] | LBW | social | individual | maternal health | CA |
| 2005 | Lasker et al. 2005 [18] | LBW | social | individual | maternal health | PA |
| 2008 | Grady et al. 2008 [174] | LBW | social | individual | maternal health | NY |
| 2013 | Heaman et al. 2013 [175] | SGA | social | individual | maternal health | Canada |
| 2014 | Aris et al. 2014 [176] | LBW, IUGR | social | individual | endometriosis | QC |
| 2015 | Chen et al. 2015 [177] | LBW, SGA | social | individual | interpregnancy interval | AB |
| 2016 | Shapiro et al. 2016 [178] | SGA | social | individual | individual | Canada |
| 2018 | Jain et al. 2018 [179] | SGA | social | individual | maternal health | NS |
| 1999 | Gorman et al. 1999 [180] | LBW | social | race | race | USA |
| 2004 | Wenman et al. 2004 [181] | LBW | social | race | race | AB |
| 2008 | Vinikoor et al. 2008 [182] | LBW | social | race | race | NC |
| 2009 | Reichman et al. 2009 [183] | BW | social | race | race | CA, TX, MD, MI, NJ, PA, VA, IN, WI, NY, MA, TN, IL, FL, OH, NM |
| 2010 | Grady et al. 2010 [184] | IUGR | social | race | race | MI |

**Table A2.** *Cont.*

| Year | Study | Outcome [1] | Environment | Category | Variable(s) | Geography [2] |
|---|---|---|---|---|---|---|
| 2010 | Nepomnyaschy et al. 2010 [185] | LBW | social | race | race | USA |
| 2011 | Anthopolos et al. 2011 [186] | LBW, BW | social | race | race | NC |
| 2011 | Kirby et al. 2011 [187] | LBW | social | race | race | GA, SC |
| 2013 | Wallace et al. 2013 [188] | LBW | social | race | race | LA |
| 2016 | Oster et al. 2016 [189] | LBW | social | race | race | AB |
| 2018 | Shapiro et al. 2018 [190] | SGA | social | race | race | Canada |
| 1993 | Kieffer et al. 1993 [191] | LBW | social | SES | SES | HI |
| 2003 | Krieger et al. 2003 [192] | LBW | social | SES | SES, blood Pb | MA, RI |
| 2006 | Farley et al. 2006 [193] | IUGR | social | SES | SES | LA |
| 2007 | Masi et al. 2007 [194] | BW | social | SES | SES, built | IL |
| 2008 | Zeka et al. 2008 [195] | SGA, BW | social | SES | SES, built | MA |
| 2010 | Young et al. 2010 [196] | BW | social | SES | SES | MA |
| 2012 | Tu et al. 2012 [197] | BW | social | SES | SES | GA |
| 2013 | Auger et al. 2013 [198] | SGA | social | SES | SES | QC |
| 2013 | Legerski et al. 2013 [199] | LBW | social | SES | SES | KS |
| 2013 | Meng et al. 2013 [200] | LBW | social | SES | SES | ON |
| 2015 | Chan et al. 2015 [201] | LBW, SGA | social | SES | SES | Canada |
| 2015 | Shmool et al. 2015 [202] | BW | social | SES | SES, $NO_2$ | NY |
| 2016 | Martinson et al. 2016 [45] | LBW | social | SES | SES | Canada, USA |
| 2017 | Bushnik et al. 2017 [203] | SGA | social | SES | SES | Canada |
| 2017 | MacQuillan et al. 2017 [204] | LBW | social | SES | SES | MI |
| 2018 | Campbell et al. 2018 [205] | LBW | social | SES | SES | ON |
| 2018 | McRae et al. 2018 [206] | SGA | social | SES | SES | BC |

[1] Outcomes included low birth weight (LBW), small for gestational age (SGA), birth weight (BW), and intrauterine growth restriction (IUGR). [2] Geography abbreviations are detailed in Table A1.

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
