# Peer review of "Geographical Analysis of the Distribution of Publications Describing Spatial Associations among Outdoor Environmental Variables and Really Small Newborns in the USA and Canada"

_challenges, doi:10.3390/challe10010011_

Round 1

Reviewer 1 Report

Lines 30-31: add Canada and USA in your keyswords

Lines 65-67: caution in the use of agregated terms like Low-and-middle income countries, as this classification hides many disparities. If possible, disagregate by referring either to Subsaharan africa countries, or to Central and south America countries, or again to South east Asian countries

Lines 108-121: This paragraph seems useless. Either delete or synthetize

Line 152: By restricting your bibliograhic search to english-speaking authors, you introduce a conceptual bias. There are french, portuguese and ...speaking researchers who have extensively authored original studies and that could be highlly useful

Author Response

We appreciate your thoughtful review and have addressed all your suggestions (please see in-track changes):

added Canada and USA to the keywords (line 31)

highlighted Africa and east Asia in the global statistics (lines 66 and 68)

this places our work in the context of planetary health - the subject of the journal's special issue - and wish to keep it as is

added discussion on the conceptual bias to the discussion section along with an additional comment by another reviewer (lines 267-270)

Thank you!

Reviewer 2 Report

The article gives us a broad vision of the possible relationships existing between inadequate outdoor environment and low birth weight. Even though, at the geographical scale used, it is not possible to point a cause effect relation, it may help set new research hypothesis and serve as basis for new investigations and for public policies. 

Another richness of the manuscript is to bring together the most up to date literature on the theme published in English language.

The methods are well described and are adequate for a mapping literature review.

The illustrations are fine and support the results.

The conclusions are adequate and based on the findings.

Author Response

We appreciate your thoughtful feedback.

We liked and added in your comment about up-to-date literature within another reviewer's suggestion (lines 267-270).

Thank you.